# Fascial Signal Change on the Cervical MRI of a Patient with Systemic Lupus Erythematosus

**DOI:** 10.3390/diagnostics14010010

**Published:** 2023-12-20

**Authors:** Hyun-Je Kim, Dong Gyu Lee

**Affiliations:** 1Division of Rheumatology, Department of Internal Medicine, Yeungnam University College of Medicine, Daegu 42415, Republic of Korea; hjkim23@yu.ac.kr; 2Department of Physical Medicine and Rehabilitation, Yeungnam University College of Medicine, Daegu 42415, Republic of Korea

**Keywords:** systemic inflammation, fat suppression view, PET-CT, systemic lupus erythematosus

## Abstract

Here, we present a case of a 53-year-old female patient with chronic neck pain and systemic inflammation who was ultimately diagnosed with systemic lupus erythematosus. Notably, applying fat-suppressed T2-weighted MRI sequences was pivotal in detecting structural fascial changes commonly associated with systemic inflammatory diseases. PET-CT scans further revealed systemic inflammation around multiple joints, providing valuable insights into MRI signal alterations. This case underscores the importance of considering systemic autoimmune pathology as a potential underlying cause of chronic musculoskeletal pain. It also highlights the clinical utility of MRI with fat suppression sequences in identifying inflammation-related fascial changes. This case emphasizes the significance of a comprehensive evaluation, particularly in situations where clinical features overlap between autoimmune and degenerative skeletal pathologies. Fat-suppressed MRI can provide information about fascial pathology related to systemic inflammatory diseases. In this context, it is worth noting that PET-CT and fat suppression MRI complement each other by providing complementary information about inflammation and the underlying causes of a patient’s pain.

A 53-year-old female patient who complained of posterior neck pain and numbness in both arms visited the Spine Center’s outpatient clinic. The patient reported experiencing pain for the past 8 months, with a worsening of symptoms, especially after receiving chiropractic therapy. There were no specific underlying medical conditions. The patient complained of morning pain and persistent pain during the day. Although there were no abnormal findings in the physical examination, she reported pain as moving her shoulders. She had previously received treatment with medication, physical therapy, and injections at several hospitals, but her pain did not improve, leading to a referral to our hospital. The patient was admitted for a differential diagnosis.

The patient’s blood tests, including inflammatory markers, yielded the following values: white blood cell (WBC) count of 2850/µL, hemoglobin (Hb) level of 10.4 g/dL, platelet (Plt) count of 351 × 10^3^/µL, C-reactive protein (CRP) level of 3.46 mg/dL, and erythrocyte sedimentation rate (ESR) level of 15 mm/h. The patient experienced febrile sensations with temperatures reaching up to 39.1 °C for five days. As per the evaluation of a fever of unknown origin (FUO), the patient underwent PET-CT, autoantibodies, and immunological tests. The results indicated a positive FANA (Fluorescent Antinuclear Antibody) test with a titer of 1:1280, a positive result of anti-ds-DNA IgG (Anti-Double-Stranded DNA Immunoglobulin G) at 48.9 IU/mL (reference range: 0~20 IU/mL), and decreased complement 3(C3) level of 71.9 mg/dL (reference range: 90~180 mg/dL). The rheumatoid factor and Anti CCP antibody tests yielded negative results. The patient reported swelling and stiffness in the joints from both the 2nd to the 4th proximal interphalangeal joints (PIPJs) and from the 2nd to the 5th metacarpophalangeal joints (MCPJs) and in both wrist joints, with corresponding FDG uptakes observed on the PET-CT scan. Each pleural effusion was also observed during the chest CT, without evidence of bacterial infection, including tuberculosis and malignancy, in a pleural fluid study. Based on the 2019 European League Against Rheumatism (EULAR)/American College of Rheumatology (ACR) classification criteria [1], the patient was diagnosed with systemic lupus erythematosus (SLE). The SLE activity assessed using the SLE Disease Activity Index 2000 (SLEDAI 2K) was 12 [2]. The patient was treated with hydroxychloroquine and prednisolone, which led to a gradual improvement in both inflammation and pain.

A C-spine MRI was performed, revealing evidence of degenerative spondylosis at the C5-6 level (Figure 1). The findings suggested potential causes for the patient’s pain, as considered by another hospital. However, the MRI also showed signal changes in the prevertebral area and the intermuscular fascia of the posterior neck area. While signal changes in such soft tissues are commonly observed in cases of traumatic injury, they are not frequently seen in cases of non-traumatic degeneration [3]. Inflammation in soft tissues is often observed in autoimmune diseases such as rheumatologic diseases and vasculitis [4]. SLE activates immune cells through nuclear self-antigens, inducing inflammation in connective tissues such as cartilage, ligaments, muscles, skin, blood vessels, and other soft tissues. The fascia is a tissue that encompasses these connective tissues, including interosseous membranes, tendons, entheses, epimysium, and vessels.

The fat-suppressed T2-weighted image is a sensitive MRI sequence for confirming active enthesitis [5]. Furthermore, the intermuscular fascia, which contains a significant amount of fatty tissue, may not display abnormalities in T2-weighted images without fat suppression. In this patient, the T2-weighted images did not reveal any specific findings in the posterior cervical area, but significant signal changes were observed in the fat-suppressed T2-weighted view. Therefor, for diffential diagnosis, the fat suppression view provides valuable information. Similarly, this case demonstrates the importance of fat-suppressed T2-weighted images in identifying structural changes in fascia, which are frequently affected in systemic inflammatory diseases.

Although the PET-CT scan in this patient showed no evidence of FDG uptake in the prevertebral area and posterior neck fascia, it did reveal FDG uptake around multiple joints. (Figure 2) This suggests that this patient had systemic inflammation, which provides a possible explanation for the MRI signal changes. The fascia is a large complex of connective tissue found throughout the body. It contains nociceptors, meaning that inflammatory processes can lead to localized pain [6]. Additionally, many nerves traverse through the fascia to reach their respective locations; so, inflammation-induced changes in the extracellular matrix (ECM) can increase fascial stiffness [7]. These alterations can influence fascial nociception and stimulate nerves through stiffness, potentially causing musculoskeletal pains. So, hydrodilation procedures are performed to release peripheral nerve compression caused by stiff fascia [8]. The deep cervical fascia, palmar fascia, iliotibial band, and thoracolumbar fascia are known fasciae considered as potential causes of musculoskeletal pain [9]. While research on neck fascia is limited compared to other areas, previous study have shown increased fascia thickness in the SCM (sternocleidomastoid) and middle scalene muscles in patients with chronic neck pain [10]. Additionally, fascial manipulation reduced fascia thickness and pain.

This case illustrates that systemic autoimmune pathology can present clinical features like degenerative spinal pathology. Furthermore, it demonstrates the clinical utility of fat suppression sequences in spinal MRI examinations, especially when there is fascial signal change, highlighting the need for careful consideration of the possibility of systemic autoimmune pathology.

## Figures and Tables

**Figure 1 diagnostics-14-00010-f001:**
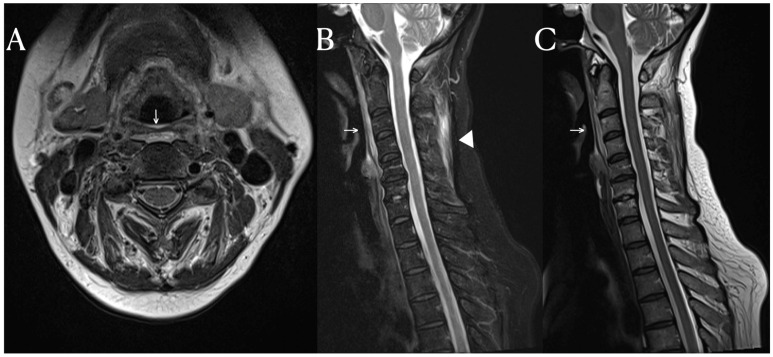
Cervical spine MRI. In the T2-weighted sagittal image, mild edema and high signal changes in the prevertebral soft tissue (arrow) are found (**A**–**C**). Notably, on the fat suppression T2-weighted image, additional findings of high signal intensity in the posterior neck fascia (arrowhead) are observed (**B**).

**Figure 2 diagnostics-14-00010-f002:**
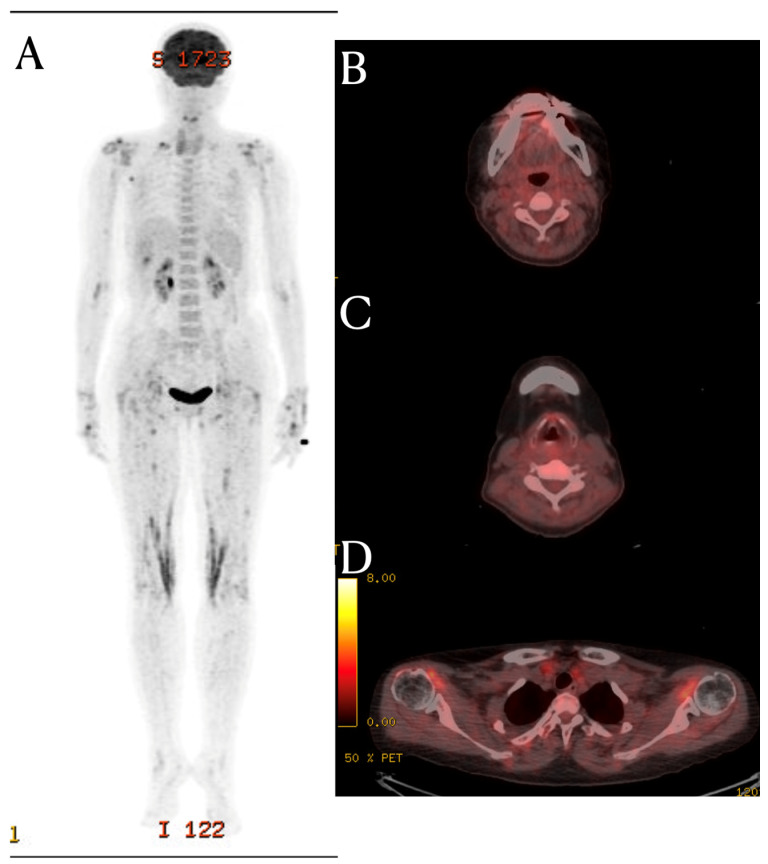
FDG PET-CT finding. Bilateral shoulder shows uneven FDG uptake (**D**). Both wrists, hips, and knees show increased FDG uptake (**A**). However, in the FDG PET-CT images of the C3 (**B**) and C5 spinal level (**C**), there is no hot uptake in the prevertebral soft tissue and posterior neck fascia areas, where signal changes were seen in the MRI.

## Data Availability

Data are available upon reasonable request from the authors.

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
