# Peer review of "Fascial Signal Change on the Cervical MRI of a Patient with Systemic Lupus Erythematosus"

_diagnostics, 2023, doi:10.3390/diagnostics14010010_

Round 1

Reviewer 1 Report

Comments and Suggestions for Authors

This manuscript seems to be an interesting case report of a patient whose fat-suppressed T2-weighted MR imaging revealed fascial pathology related with systemic inflammatory diseases, not depicted on FDG PET-CT.

It is recommended to show an FDG PET-CT image, with negative findings, corresponding to those with sagittal cervical MR images shown in Figure 1.

Additionally, if the authors performed diffusion-weighted imaging and/or post-contrast T1-weighted imaging, it is advisable to add the findings.

In other parts, this manuscript can be published as it is.

Author Response

We appreciate the insightful and helpful comments of the editor and reviewers very much. We have made as many changes as possible according to the editor’s and reviewer’s recommendations and have prepared the responses in a point-by-point fashion. We hope that our revision is satisfactory to the standards of the editors and reviewers and look forward to hearing the ultimate decision.

This manuscript seems to be an interesting case report of a patient whose fat-suppressed T2-weighted MR imaging revealed fascial pathology related with systemic inflammatory diseases, not depicted on FDG PET-CT.

It is recommended to show an FDG PET-CT image, with negative findings, corresponding to those with sagittal cervical MR images shown in Figure 1.

Respond: In response to the suggestion to include an FDG PET-CT image corresponding to the negative findings in sagittal cervical MR images shown in Figure 1, we added a PET-CT axial view in Figure 2, corresponding to the relevant area. Figures 2B and 2C in Figure 2 demonstrate the absence of hot uptake on FDG PET-CT, corresponding to the signal changes and swelling observed in MRI at C3-5.

Additionally, if the authors performed diffusion-weighted imaging and/or post-contrast T1-weighted imaging, it is advisable to add the findings.

Respond: Unfortunately, we did not conduct a diffusion-weighted or post-contrast imaging study.

In other parts, this manuscript can be published as it is.

Reviewer 2 Report

Comments and Suggestions for Authors

The article presents a SLE case with systemic inflammation on MRI. It is well written, yet I recommend taking into consideration the following remarks:

·        provide the activity score also (SLEDAI-score or other) if there is any available 

·        R59: add some information about the process of inflammation in soft tissues in autoimmune diseases, with a focus on SLE

·        R 65-73 rephrase or change the order of the phrases inside the paragraph 

·        I suggest including a future perspective of the findings you have presented to make it more interesting (a clinical perspective, treatment options or change in the prognosis of the patient)

Author Response

The authors appreciated the Reviewer’s comments for contributing to the improvement of this article.

The article presents a SLE case with systemic inflammation on MRI. It is well written, yet I recommend taking into consideration the following remarks:

provide the activity score also (SLEDAI-score or other) if there is any available 

Respond: Considering the patient’s symptoms and laboratory results, the SLE disease activity according to SLEDAI 2000 is 12 points (4 points for arthritis (2 or more joints with pain and signs of inflammation) + 2 points for pleuritic chest pain with pleural effusion + 2 points for low complements + 2 points for increased DNA binding (Anti-dsDNA IgG positive) + 1 point for body temperature > 38C, excluding infectious cause + 1 point for leukopenia, WBC <3 x 10^9/L). The authors have revised the sentences based on the Reviewer’s comments in lines 51-52.

R59: add some information about the process of inflammation in soft tissues in autoimmune diseases, with a focus on SLE

Respond: As mentioned, we have added information about SLE and soft tissue inflammation.

R 65-73 rephrase or change the order of the phrases inside the paragraph 

Respond: We have reorganized sentences 65-73 within the paragraph as per the reviewer's comment.

I suggest including a future perspective of the findings you have presented to make it more interesting (a clinical perspective, treatment options or change in the prognosis of the patient)